# Inferring collective dynamical states from widely unobserved systems

Jens Wilting[1] & Viola Priesemann [1,2]

When assessing spatially extended complex systems, one can rarely sample the states of all components. We show that this spatial subsampling typically leads to severe under-estimation of the risk of instability in systems with propagating events. We derive a subsampling-invariant estimator, and demonstrate that it correctly infers the infectiousness of various diseases under subsampling, making it particularly useful in countries with unreliable case reports. In neuroscience, recordings are strongly limited by subsampling. Here, the subsampling-invariant estimator allows to revisit two prominent hypotheses about the brain's collective spiking dynamics: asynchronous-irregular or critical. We identify consistently for rat, cat, and monkey a state that combines features of both and allows input to reverberate in the network for hundreds of milliseconds. Overall, owing to its ready applicability, the novel estimator paves the way to novel insight for the study of spatially extended dynamical systems.

[1] Max-Planck-Institute for Dynamics and Self-Organization, Am Faßberg 17, 37077 Göttingen, Germany. [2] Bernstein-Center for Computational Neuroscience, Am Faßberg 17, 37077 Göttingen, Germany. These authors contributed equally: Jens Wilting, Viola Priesemann. Correspondence and requests for materials should be addressed to V.P. (email: viola@nld.ds.mpg.de)

How can we infer properties of a high-dimensional dynamical system if we can only observe a very small part of it? This problem of spatial subsampling is common to almost every area of research where spatially extended, time evolving systems are investigated. For example, in many diseases the number of reported infections may be much lower than the unreported ones[1], or in the financial system only a subset of all banks is evaluated when assessing the risk of developing system wide instability[2] ("stress test"). Spatial subsampling is particularly severe when recording neuronal spiking activity, because the number of neurons that can be recorded with millisecond precision is vanishingly small compared to the number of all neurons in a brain area[3–5].

Here, we show that subsampling leads to a strong overestimation of stability in a large class of time evolving systems, which include epidemic spread of infectious diseases[6], cell proliferation, evolution (see ref. [7] and references therein), neutron processes in nuclear power reactors[8], spread of bankruptcy[9], evolution of stock prices[10], or the propagation of spiking activity in neural networks[11,12]. However, correct risk prediction is essential to timely initiate counter actions to mitigate the propagation of events. We introduce a novel estimator that allows correct risk assessment even under strong subsampling. Mathematically, the evolution of all these systems is often approximated by a process with a 1st order autoregressive representation (PAR), e.g., by an AR(1), branching, or Kesten process. For these processes, we derive first the origin of the estimation bias and develop a novel estimator, which we analytically prove to be consistent under subsampling. We then apply the novel estimator to models and real-world data of disease and brain activity. To assure that a PAR is a reasonable approximation of the complex system under study, and to exclude contamination through potential non-stationarities, we included a set of automated, data-driven tests.

## Results

In a PAR (Supplementary Notes 1–4), the activity in the next time step, $A_{t+1}$, depends linearly on the current activity $A_t$. In addition, it incorporates external input, e.g., drive from stimuli or other brain areas, with a mean rate $h$, yielding the autoregressive representation

$$\langle A_{t+1}|A_t \rangle = mA_t + h, \tag{1}$$

where $\langle \cdot | \cdot \rangle$ denotes the conditional expectation. The stability of $A_t$ is solely governed by $m$, e.g., the mean number of persons infected by one diseased person[13]. The activity is stationary if $m < 1$, while it grows exponentially if $m > 1$. The state $m = 1$ separates the stable from the unstable regime. Especially close to this transition, a correct estimate of $m$ is vital to assess the risk that $A_t$ develops a large, potentially devastating cascade or avalanche of events (e.g., an epidemic disease outbreak or an epileptic seizure), either generically or via a minor increase in $m$.

A conventional estimator[14,15] $\hat{m}_C$ of $m$ uses linear regression of activity at time $t$ and $t + 1$, because the slope of linear regression directly returns $m$ owing to the autoregressive representation in Eq. (1). This estimation of $m$ is consistent if the full activity $A_t$ is known. However, under subsampling it can be strongly biased, as we show here. To derive the bias quantitatively, we model subsampling in a generic manner in our stochastic framework: we assume only that the subsampled activity $a_t$ is a random variable that in expectation it is proportional to $A_t$, $\langle a_t|A_t \rangle = \alpha A_t + \beta$ with two constants $\alpha$ and $\beta$ (Supplementary Note 3). This represents, for example, sampling a fraction $\alpha$ of all neurons in a brain area. Then the conventional estimator is biased by $m(\alpha^2 \text{Var}[A_t]/\text{Var}[a_t] - 1)$ (Supplementary Corollary 6). The bias vanishes

only when all units are sampled ($\alpha = 1$, Fig. 1c–e), but is inherent to subsampling and cannot be overcome by obtaining longer recordings.

Kalman filtering[16–18], a state-of-the-art approach for system identification, cannot overcome the subsampling bias either, because it assumes Gaussian noise for both the evolution of $A_t$ and the sampling process for generating $a_t$ (Supplementary Note 7). These assumptions are violated under typical subsampling conditions, when the values of $a_t$ become too small, so that the central limit theorem is not applicable, and hence Kalman filtering fails (Fig. 1d). It is thus applicable to a much narrower set of subsampling problems and in addition requires orders of magnitude longer runtime compared to our novel estimator (Supplementary Fig. 7).

Our novel estimator takes a different approach than the other estimators (Supplementary Note 4). Instead of directly using the biased regression of activity at time $t$ and $t + 1$, we perform multiple linear regressions of activity between times $t$ and $t + k$ with different time lags $k = 1, \ldots, k_{\max}$. These return a collection of linear regression slopes $r_k$ (note that $r_1$ is simply the conventional estimator $\hat{m}_C$). Under full sampling, one expects an exponential relation[19] $r_k = m^k$ (Supplementary Theorem 2). Under subsampling, however, we showed that all regressions slopes $r_k$ between $a_t$ and $a_{t+k}$ are biased by the same factor $b = \alpha^2 \text{Var}[A_t]/\text{Var}[a_t]$ (Supplementary Theorem 5). Hence, the exponential relation generalizes to

$$r_k = \alpha^2 \frac{\text{Var}[A_t]}{\text{Var}[a_t]} m^k = bm^k \tag{2}$$

under subsampling. The factor $b$ is, in general, not known and thus $m$ cannot be estimated from any $r_k$ alone. However, because $b$ is constant, one does not need to know $b$ to estimate $\hat{m}$ from regressing the collection of slopes $r_k$ against the exponential model $bm^k$ according to Eq. (2). This result serves as the heart of our new multiple-regression (MR) estimator (Fig. 1f, Supplementary Figs. 1 and 2, Supplementary Corollary 3).

In fact, MR estimation is equivalent to estimating the autocorrelation time of subcritical PARs, where autocorrelation and regression $r_k$ are equal: we showed that subsampling decreases the autocorrelation strength $r_k$, but the autocorrelation time $\tau$ is preserved. This is because the system itself evolves independently of the sampling process. While subsampling biases each regression $r_k$ by decreasing the mutual dependence between subsequent observations ($a_t$, $a_{t+k}$), the temporal decay in $r_k \sim m^k = e^{-k\Delta t/\tau}$ remains unaffected, allowing for a consistent estimate of $m$ even when sampling only a single unit (Fig. 1d). Here, $\tau = -\Delta t/\log m$ refers to the autocorrelation time of stationary (subcritical) processes, where autocorrelation and regression $r_k$ are equal, and $\Delta t$ is the time scale of the investigated process. Particularly close to $m = 1$ the autocorrelation time $\tau = -\Delta t/\log m$ diverges, which is known as critical slowing down[20]. Because of this divergence, MR estimation can resolve the distance to criticality in this regime with high precision. Making use of this result allows for a consistent estimate of $m$ even when sampling only a single unit (Fig. 1d).

PARs are typically only a first order approximation of real world event propagation. However, their mathematical structure allowed for an analytical derivation of the subsampling bias and the consistent estimator. To show that the MR estimator returns correct results also for more complex systems, we applied it to more complex simulated systems: a branching network[12] (BN) and the non-linear Bak–Tang–Wiesenfeld model[21] (BTW, see Supplementary Note 8). In contrast to generic PARs, these models (a) run on recurrent networks and (b) are of finite size. In addition, the second model shows (c) completely deterministic

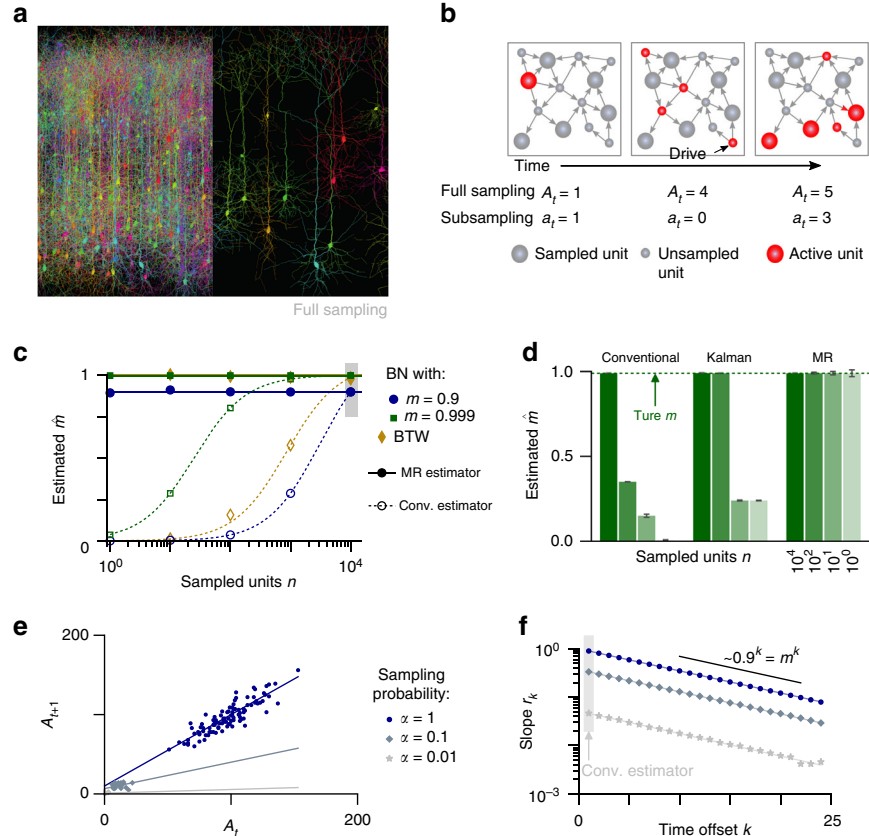

**Fig. 1** Spatial subsampling. **a** In complex networks, such as the brain, often only a small subset of all units can be sampled (spatial subsampling); figure created using TREES[57]. **b** In a branching network (BN), an active unit (e.g., a spiking neuron, infected individual, or defaulting bank) activates some of its neighbors in the next time step. Thereby activity can spread over the system. Units can also be activated by external drive. As the subsampled activity $a_t$ may significantly differ from the actual activity $A_t$, spatial subsampling can impair inferences about the dynamical properties of the full system. **c** In recurrent networks (BN, Bak-Tang-Wiesenfeld model (BTW)), the conventional estimator (empty symbols) substantially underestimates the branching ratio $m$ when less units $n$ are sampled, as theoretically predicted (dashed lines). The novel multistep regression (MR) estimator (full symbols) always returns the correct estimate, even when sampling only 10 or 1 out of all $N = 10^4$ units. **d** For a BN with $m = 0.99$, the conventional estimator infers $\hat{m} = 0.37$, $\hat{m} = 0.1$, or $\hat{m} = 0.02$ when sampling 100, 10, or 1 units, respectively. Kalman filtering based estimation returns approximately correct values under slight subsampling ($n = 100$), but is biased under strong subsampling. In contrast, MR estimation returns the correct $\hat{m}$ for any subsampling. **e** MR estimation is exemplified for a subcritical branching process ($m = 0.9$, $h = 10$), where active units are observed with probability $\alpha$. Under subsampling (gray), the regression slopes $r_1$ are smaller than under full sampling (blue). **f** While conventional estimation of $m$ relies on the linear regression $r_1$ and is biased under subsampling, MR estimation infers $\hat{m}$ from the exponential relation $r_k \propto m^k$, which remains invariant under subsampling

propagation of activity instead of the stochastic propagation that characterizes PARs, and (d) the activity of each unit depends on many past time steps, not only one. Both models approximate neural activity propagation in cortex[3,4,11,12,22,23]. For both models the numerical estimates of $m$ were precisely biased as analytically predicted, although the models are only approximated by a PAR (dashed lines in Fig. 1c, Supplementary Eq. (4)). The bias is considerable: for example, sampling 10% or 1% of the neurons in a BN with $m = 0.9$ resulted in the estimates $\hat{m}_C = r_1 = 0.312$, or even $\hat{m}_C = 0.047$, respectively. Thus a process fairly close to instability ($m = 0.9$) is mistaken as Poisson-like ($\hat{m}_C = 0.047 \approx 0$) just because sampling is constrained to 1% of the units. Thereby the risk that systems may develop instabilities is severely underestimated.

MR estimation is readily applicable to subsampled data, because it only requires a sufficiently long time series $a_t$, and the assumption that in expectation $a_t$ is proportional to $A_t$. Hence, in general it suffices to sample the system randomly, without even knowing the system size $N$, the number of sampled units $n$, or any moments of the underlying process. Importantly, one can obtain a consistent estimate of $m$, even when sampling only a very small fraction of the system, under homogeneity even when sampling

only one single unit (Fig. 1c, d, Supplementary Fig. 6). This robustness makes the estimator readily applicable to any system that can be approximated by a PAR. We demonstrate the bias of conventional estimation and the robustness of MR estimation at the example of two real-world applications.

**Application to disease case reports**. We used the MR estimator to infer the "reproductive number" $m$ from incidence time series of different diseases[24]. Disease propagation represents a non-linear, complex, real-world system often approximated by a PAR[25,26]. Here, $m$ determines the disease spreading behavior and has been deployed to predict the risk of epidemic outbreaks[6]. However, the problem of subsampling or under-ascertainment has always posed a challenge[1,27].

As a first step, we cross-validated the novel against the conventional estimator using the spread of measles in Germany, surveyed by the Robert-Koch-Institute (RKI). We chose this reference case, because we expected case reports to be almost fully sampled owing to the strict reporting policy supported by child care facilities and schools[28,29], and to the clarity of symptoms. Indeed, the values for $\hat{m}$ inferred with the conventional and with

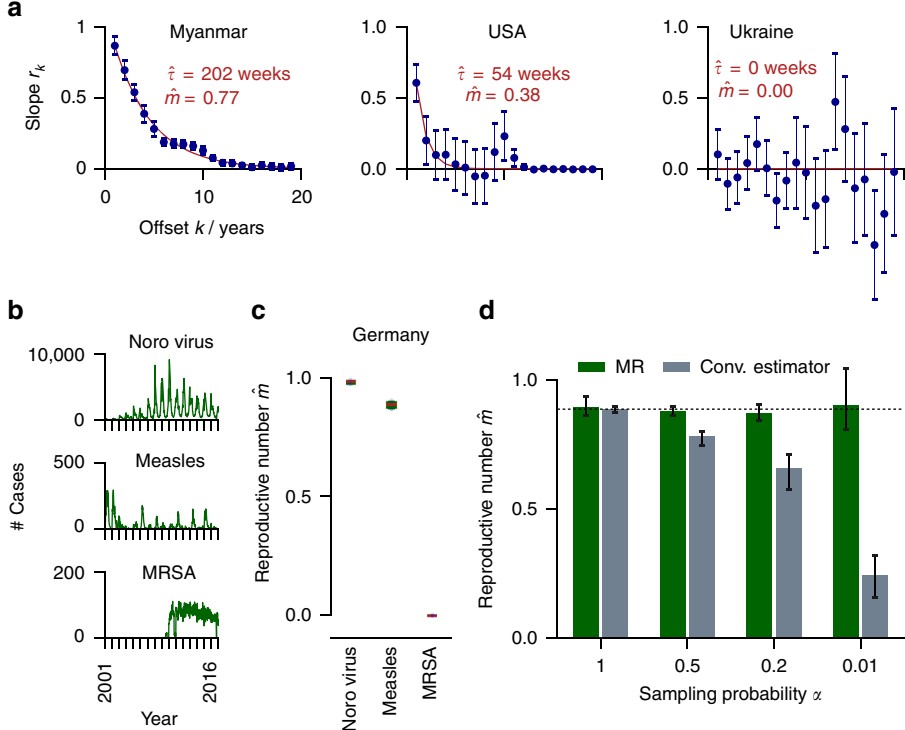

**Fig. 2** Disease propagation. In epidemic models, the reproductive number $m$ can serve as an indicator for the infectiousness of a disease within a population, and predict the risk of large incidence bursts. We have estimated $\hat{m}$ from incidence time series of measles infections for 124 countries worldwide (Supplementary Note 9); as well as noroviral infection, measles, and invasive meticillin-resistant *Staphylococcus aureus* (MRSA) infections in Germany. **a** MR estimation of $\hat{m}$ is shown for measles infections in three different countries. Error bars here and in all following figures indicate 1SD or the corresponding 16 to 84% confidence intervals if asymmetric. The reproductive numbers $\hat{m}$ decrease with the vaccination rate (Spearman rank correlation: $r = -0.342$, $p < 10^{-4}$). **b** Weekly case report time series for norovirus, measles and MRSA in Germany. **c** Reproductive numbers $\hat{m}$ for these infections. **d** When artificially subsampling the measles recording (under-ascertainment), conventional estimation underestimates $\hat{m}_C$, while MR estimation still returns the correct value. Both estimators return the same $\hat{m}$ under full sampling

the novel estimator, coincided (Fig. 2d, Supplementary Note 9). In contrast, after applying artificial subsampling to the case reports, thereby mimicking that each infection was only diagnosed and reported with probability $\alpha < 1$, the conventional estimator severely underestimated the spreading behavior, while MR estimation always returned consistent values (Fig. 2d). This shows that the MR estimator correctly infers the reproductive number $m$ directly from subsampled time series, without the need to know the degree of under-ascertainment $\alpha$.

Second, we evaluated worldwide measles case and vaccination reports for 124 countries provided by the WHO since 1980 (Fig. 2a, Supplementary Note 9), because the vaccination percentage differs in each country, and this is expected to impact the spreading behavior through $m$. The reproductive numbers $\hat{m}$ ranged between 0 and 0.93, and in line with our prediction clearly decreased with increasing vaccination percentage in the respective country (Spearman rank correlation: $r = -0.342$, $p < 10^{-4}$).

Third, we estimated the reproductive numbers for three diseases in Germany with highly different infectiousness: noroviral infection[27,30], measles, and invasive meticillin-resistant *Staphylococcus aureus* (MRSA, an antibiotic-resistant germ classically associated with health care facilities[31], Fig. 2b, c), and quantified their propagation behavior. MR estimation returned the highest $\hat{m} = 0.98$ for norovirus, compliant with its high infectiousness[32]. For measles we found the intermediate $\hat{m} = 0.88$, reflecting the vaccination rate of about 97%. For MRSA we identified $m = 0$, confirming that transmission is still minor in Germany[33]. However, a future increase of transmission is feared and would pose a major public health risk[34]. Such an

increase could be detected by our estimator, even in countries where case reports are incomplete.

**Reverberating spiking activity in vivo**. We applied the MR estimator to cortical spiking activity in vivo to investigate two contradictory hypotheses about collective spiking dynamics. One hypothesis suggests that the collective dynamics is "asynchronous irregular" (AI)[35–38], i.e., neurons spike independently of each other and in a Poisson manner ($m = 0$), which may reflect a balanced state[39–41]. The other hypothesis suggests that neuronal networks operate at criticality ($m = 1$)[3,11,42–44], thus in a particularly sensitive state close to a phase transition. These different hypotheses have distinct implications for the coding strategy of the brain: Criticality is characterized by long-range correlations in space and time, and in models optimizes performance in tasks that profit from long reverberation of the activity in the network[12,45–48]. In contrast, the typical balanced state minimizes redundancy[49] and supports fast network responses[39].

Analyzing in vivo spiking activity from Macaque monkey prefrontal cortex during a memory task, anesthetized cat visual cortex with no stimulus (Fig. 3a, b), and rat hippocampus during a foraging task (Supplementary Note 10) returned $\hat{m}$ to be between 0.963 and 0.998 (median $\hat{m} = 0.984$, Fig. 3e, Supplementary Fig. 5), corresponding to autocorrelation times between 100 and 2000 ms. This clearly suggests that spiking activity in vivo is neither AI-like ($m = 0$), nor consistent with a critical state ($m = 1$), but in a reverberating state that shows autocorrelation times of a few hundred milliseconds. We call the range of the dynamical states found in vivo reverberating, because input

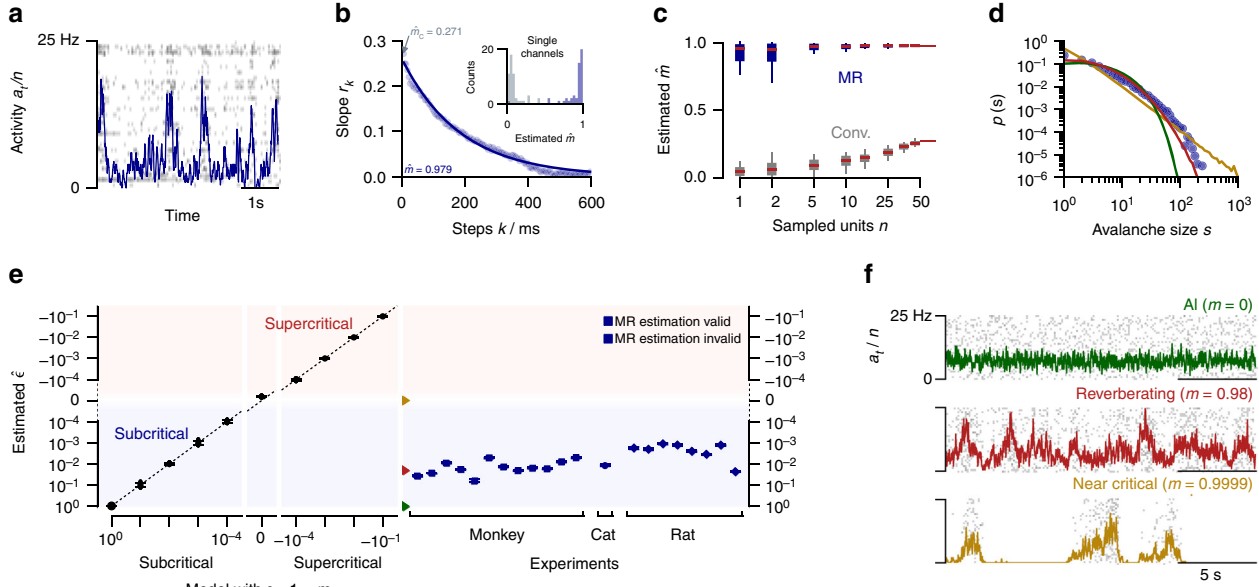

**Fig. 3** Animal spiking activity in vivo. In neuroscience, $m$ denotes the mean number of spikes triggered by one spike. We estimated $\hat{m}$ from spiking activity recorded in vivo in monkey prefrontal cortex, cat visual cortex, and rat hippocampus. **a** Raster spike plot and population rate $a_t$ of 50 single units illustrated for cat visual cortex. **b** MR estimation based on the exponential decay of the autocorrelation $r_k$ of $a_t$. Inset: Comparison of conventional and MR estimation results for single units (medians $\hat{m}_C = 0.057$ and $\hat{m} = 0.954$, respectively). **c** $\hat{m}$ estimated from further subsampled cat recordings, estimated with the conventional and MR estimator. Error bars indicate variability over 50 randomly subsampled $n$ out of the recorded 50 channels. **d** Avalanche size distributions for cat visual cortex (blue) and the networks with AI, reverberating and near-critical dynamics in **f**. **e** For all simulations, MR estimation returned the correct distance to instability (criticality) $\epsilon = 1 - m$ (Supplementary Note 8). In vivo spike recordings from rat, cat, and monkey, clearly differed from critical ($\epsilon = 0$) and AI ($\epsilon = 1$) states (median $\hat{m} = 0.98$, error bars: 16 to 84% confidence intervals, note that some confidence intervals are too small to be resolved). Opaque symbols indicate that MR estimation was rejected (Supplementary Fig. 5, Supplementary Note 5). Green, red, and yellow arrows indicate $\epsilon$ for the dynamic states shown in **f**. **f** Population activity and raster plots for AI activity, reverberating, and near critical networks. All three networks match the recording from cat visual cortex with respect to number of recorded neurons and mean firing rate

reverberates for a few hundred millisecond in the network, and therefore enables integration of information[50–52]. Thereby the reverberating state constitutes a specific narrow window between AI state, where perturbations of the firing rate are quenched immediately, and the critical state, in which perturbations can in principle persist infinitely long (for more details, see Wilting and Priesemann[53]).

We demonstrate the robustness to subsampling for the activity in cat visual cortex: we chose random subsets of $n$ neurons from the total of 50 recorded single units. For any subset, even for single neurons, MR estimation returned about the same median $\hat{m}$ (Fig. 3c). In contrast, the conventional estimator misclassified neuronal activity by strongly underestimating $\hat{m}$: instead of $\hat{m} = 0.984$, it returned $\hat{m}_C = 0.271$ for the activity of all 50 neurons. This underestimation gets even more severe when considering stronger subsampling ($n < 50$, Fig. 3c). Ultimately, for single neuron activity, the conventional estimator returned $\hat{m}_C = 0.057 \approx 0$, which would spuriously indicate dynamics close to AI instead of the reverberating state (inset of Fig. 3b, c and Supplementary Fig. 6). The underestimation of $\hat{m}_C$ was present in all experimental recordings ($r_1$ in Supplementary Fig. 5).

On first sight, $\hat{m} = 0.984$ may appear close to the critical state, particularly as physiologically a 1.6% difference to $m = 1$ is small in terms of the effective synaptic strength. However, this seemingly small difference in single unit properties has a large impact on the collective dynamics and makes AI, reverberating, and critical states clearly distinct. This distinction is readily manifest in the fluctuations of the population activity (Fig. 3f). Furthermore, the distributions of avalanche sizes clearly differ from the power-law scaling expected for critical systems[11], but are well captured by a matched, reverberating model (Fig. 3d).

Because of the large difference in the network dynamics, the MR estimator can distinguish AI, reverberating, and critical states with the necessary precision. In fact, the estimator would allow for 100 times higher precision when distinguishing critical from non-critical states, assuming in vivo-like subsampling and mean firing rate (sampling $n = 100$ from $N = 10^4$ neurons, Fig. 3e). With larger $N$, this discrimination becomes even more sensitive (detailed error estimates: Supplementary Fig. 4 and Supplementary Note 6). As the number of neurons in a given brain area is typically much higher than $N = 10^4$ in the simulation, finite size effects are not likely to account for the observed deviation from criticality $\epsilon = 1 - m \approx 10^{-2}$ in vivo, supporting that in rat, cat, and monkey the brain does not operate in a critical state. Still, additional factors like input or refractory periods may limit the maximum attainable $m$ to quasi-critical dynamics on a Widom line[54], which could in principle conform with our results.

## Discussion

Most real-world systems, including disease propagation or cortical dynamics, are more complicated than a simple PAR. For cortical dynamics, for example, heterogeneity of neuronal morphology and function, non-trivial network topology, and the complexity of neurons themselves are likely to have a profound impact onto the population dynamics[55]. In order to test for the applicability of a PAR approximation, we defined a set of conservative tests (Supplementary Note 5 and Supplementary Table 1) and included only those time series, where the approximation by a PAR was considered appropriate. For example, we excluded all recordings that showed an offset in the slopes $r_k$, because this offset is, strictly speaking, not explained by a PAR and might indicate non-stationarities (Supplementary Fig. 3).

Even with these conservative tests, we found the exponential relation $r_k = bm^k$ expected for PARs in the majority of real-world time series (Supplementary Fig. 5, Supplementary Note 9). This shows that a PAR is a reasonable approximation for dynamics as complex as cortical activity or disease propagation. With using PARs, we draw on the powerful advantage of analytical tractability, which allowed for valuable insight into dynamics and stability of the respective system. It is then a logical next step to refine the model by including additional relevant parameters[56]. However, the increasing richness of detail typically comes at the expense of analytical tractability.

By employing for the first time a consistent, quantitative estimation, we provided evidence that in vivo spiking population dynamics reflects a stable, fading reverberation state around $m = 0.98$ universally across different species, brain areas, and cognitive states. Because of its broad applicability, we expect that besides the questions investigated here, MR estimation can substantially contribute to the understanding of real-world dynamical systems in diverse fields of research where subsampling prevails.

**Data availability**. Time series with yearly case reports for measles in 194 different countries are available online from the World Health Organization (WHO) for the years between 1980 and 2014. Weekly case reports for measles, norovirus, and invasive meticillin-resistant *Staphylococcus aureus* in Germany are available through their SURVSTAT@RKI server of the Robert-Koch-Institute. The data from rat hippocampus (https://doi.org/10.6080/K0Z60KZ9) and cat visual cortex (https://doi.org/10.6080/K0MW2F2J) are available from the CRCNS.org database. Python code for basic MR estimation and branching process simulation is available from github (https://github.com/jwilting/WiltingPriesemann2018). Any additional code is available from the authors upon request.

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

## Acknowledgements

We thank Matthias Munk for sharing his data. J.W. received support from the Gertrud-Reemstma-Stiftung. V.P. received financial support from the German Ministry for Education and Research (BMBF) via the Bernstein Center for Computational Neuroscience (BCCN) Göttingen under Grant No. 01GQ1005B, and by the German-Israel-Foundation (GIF) under grant number G-2391-421.13. J.W. and V.P. received financial support from the Max Planck Society.

## Author contributions

J.W. and V.P. contributed equally.

## Additional information

**Competing interests:** The authors declare no competing interests.

