## [Peer Review File · Nature Communications]

Reviewers' comments:

Reviewer #1 (Remarks to the Author):

This manuscript by Wilting and Preisemann offers a substantially improved estimate of the branching ratio for empirical data. Within the neuroscience community, the topic of criticality is very actively discussed and the last few years have seen improvements in estimating power law distributions, a potential marker for criticality. As the branching ratio is another potential marker (if $m = 1$, the system is critical), this manuscript is of similar importance. However, this manuscript offers more because it shows clearly for the first time the status of the data: these networks are actually not critical. The branching ratio, m , is slightly less than one, indicating that the collective dynamics will not show divergences as expected in a truly critical system. Thus, this manuscript is not just a new method, but a new result. I am not a specialist in epidemiology, but I could imagine that this work would offer improvements in that area as well.

The authors have written a very accessible introduction that clearly communicates the general nature of this problem. I found it easy to read, especially since the main results are reported directly and the more detailed math was placed in the supplementary information. The supplementary information will allow specialists to follow the steps in the derivation and to check the other technical points. Overall, this is very important work and I see no reason why it should not have been published already.

I read over the previous reviews and the authors' responses to them. Given that it has already been through one round of improvements, I could not find too many suggestions for it. However, I do have some points for the authors to consider:

1. Not critical and not asynchronous: The lines 204-207 seem to state that the activity is neither asynchronous intermittent (AI, $m = 0$) or critical ($m = 1$), as if in vivo activity is equally distant from both states. But a person reading this might think that it is closer to critical than to AI. Numerically this is clearly the case as $(0.98 - 0)^2$ is larger than $(0.98 - 1)^2$. It is fine to claim that it is neither, but it just seems like this deserves a bit more explanation as the numerical differences are not the same. While this is brought up again in lines 222-227, the differences in collective dynamics are not explained much, other than a qualitative figure (3e). For example, you could talk about optimality in some sense (e.g., information transmission) and measure how much that differs from a system with $m = 0.98$ from a system where $m = 0$ and a critical system where $m = 1$. By this type of measure, in vivo activity seems much closer to critical than AI, I would think. I realize that this could be a fuzzy, nonquantitative area, but effort to quantify some of the typical features advertised in critical systems could help here.

2. Spontaneous background activity: I realize that you want to keep the analysis tractable to see how far you can go, and only later add other parameters (as you explain in lines 261-264). Yet a major part of the critical/not critical debate concerns background spontaneous activity. Strictly speaking, a system can only be self-organized critical (SOC) if there is separation of timescales – the relaxation time of the system is much less than the interval between external driving events. This is unrealistic in the brain, though, as there is spontaneous (undriven) activity in neurons. The addition of spontaneous background activity can cause typical measures of criticality like the susceptibility to no longer diverge. This can make the network look slightly subcritical if viewed from the traditional branching model perspective. A different perspective would be that the network is quasi-critical and for each level of background activity there is a different optimum point. This set of points leads to a Widom line. Such a conception is different from stating that the network is slightly sub-critical. The Widom line view has a set of testable predictions that differ from those of the subcritical view. I realize that it is not the purpose of your paper to discuss this. My suggestion is that framing things merely in terms of critical/subcritical may obscure another important dimension to this research area.

Minor:

You mention the time bins that you used, and they seem reasonably motivated (the time it takes for activity to propagate from one neuron to another). Have you checked the results for robustness with respect to some variation in the bin widths? This could be relevant, as sub-sampling neurons

that are further apart should call for longer time bins, as propagation of activity takes longer for longer distances.

Very minor:

In the caption to figure 1, the term MR is used before it is explained as multistep regression three sentences later.

Line 552: date set -> data set

In supplementary section 5, you introduce the letter H to describe a set of hypotheses. This was somewhat confusing initially as you had previously used H for another purpose (line 423).

Reviewer #2 (Remarks to the Author):

Review of Wilting & Priesemann, "Inferring collective dynamical states from widely unobserved systems".

The central idea of W&P is to determine a crucial parameter " m " of an autoregressive process in an unbiased fashion even when the full activity of the system is spatially subsampled; this parameter determines if a system is strongly sub-critical, "reverberating" or near critical, or really critical. The authors show that estimating " m " by means of linear regression between two consecutive samples of temporal activity (as is claimed to be traditional) of the system is strongly biased, if the system is undersampled spatially. They propose a MR estimator that, instead of obtaining " m " from the amplitude in the system's response, looks at the decay timescale of activity. They show analytically for AR processes that a MR estimator based on the timescale is unbiased, and validate this with numerical simulations both for more complicated synthetic models as well as on real data.

I have read (and, I believe, understood) the paper. I have also read the referee reports, as I was asked to do, from when the previous manuscript was submitted to Nature. I strongly recommend this work be published in Nature Communications. Specifically, I disagree with the provided report of referee 2, who is critical about the "biological results and interpretation".

To support my argument that this work deserves publication due to its relevance to many fields, not limited to biology, it is important to make a few general statements first:

*) Neither Nature, or Nature Communications, are not exclusively biological journals — or so we are led to understand by their claims to interdisciplinarity — and therefore not every result needs to be judged just in light of "biological relevance", which Ref 2 seems to think is the ultimate publication measure. Having said that, I claim that even when the work of W&P *is* judged for biological relevance, it deserves publication (either in Nature or Nature Comms). Both disease spread and neural network activity have been studied in detail in highly sophisticated as well as simple models, and it is still debated what the data is telling us about the qualitative regime of operation of such models. For instance, in neural networks, the dispute is still ongoing about "critical" vs "asynchronous" regimes of activity. W&P give us a quantitative tool not only to decide those two (somewhat artificially imposed) categories, but discuss the whole quantitative range in between them, including the range of m close to but not equal to 1 (reverberating activity). Being a physicist, a real, biological system tuned to precisely the value of 1.0000 makes no sense for me (personally); but whether it is close to 1 or close to 0 makes sense, and W&P provide a simple, tractable tool to address this question. This is clearly biologically relevant, given the amount of published papers (also in high profile journals) hypothesizing one regime vs the other, while their statistics may fall short of correctly and convincingly identifying one or the other regime.

*) The authors do not claim that all the complex processes that motivate their question are true instances of simple autoregressive processes / branching processes, as some previous comments

of the referees seem to imply. Reality is more complicated than that, but (i) to define a “gold-standard” synthetic benchmark against which an estimator can be judged and (ii) by acknowledging that simple models often *are* used (by other researchers) to describe complex processes, sometimes due to experimental / data limitations, and so it is relevant how the parameters of those simpler models are estimated from data; (iii) to provide analytical insights and even define precisely the quantities of interest (e.g., “ m ”); it is perfectly valid that W&P treat the simple and perhaps biologically not-fully-realistic cases, such as PAR models. The paper should therefore not be judged as making claims about what mathematical model is appropriate or not for the diverse set of applications (which is anyway a research topic in its own right) but rather as discussing a statistical estimation problem in a given class of models that, if not done correctly, can lead to biologically qualitatively wrong results. It is correct to ask of authors for numerical evidence that the estimator performs well also out of the PAR-class that motivates it, and I believe they have done a good job at that.

*) One should also separate the claims to novelty of long “memory” in close-to-critical systems, and using these specific claims to derive an unbiased estimator (i.e., a statistical procedure to be applied to finite data of a real system). The fact that timescales can diverge close to criticality in many systems have been well known, and even more so than in neuroscience (that Ref 2 alludes to), this has been studied “to death” in physics, in a phenomenon known as “critical slowing down”. The authors currently phrase the paper in a statistical context (where critical slowing down might have been underappreciated?), and I would ask them, given the broad audience, to properly cite the physics literature where slowing down has been studied experimentally, theoretically, and realized to be a crucial technical obstacle when sampling (e.g., from MCMC). While the authors should acknowledge all this previous work, this doesn’t detract from their contribution, i.e., to devise a new (MR) estimator that uses the timescale of slowing down to infer “ m ”, closeness to criticality, in a way that is independent of the degree of subsampling.

*) Disclaimer: the only statement of the authors that I cannot evaluate is whether the use of autocorrelation time to infer “ m ” is really novel in the statistics literature; while I don’t know of such previous work doesn’t mean it hasn’t been done, given that my background is in physics.

Having made these remarks and while being strongly supportive of this paper, I think an interdisciplinary theoretical contribution can make the bigger impact the more clearly it is explained and the bigger audience can unambiguously understand it. To this end I ask the authors to:

1) Clarify the actual functioning of the estimator and carefully define the quantities they discuss in the main text. Specifically, the authors are unclear around lines 51/52, and also around lines 75 etc. I only understood the MR estimator after reading the SI. Specifically, what is r_1 , what is \hat{m}_C ? Also lines after 75: what is meant by “all regressions r_k ”? Is “regression” here a procedure (linear, nonlinear or whichever), or a particular estimator for the slope (i.e., not the method, but a parameter to be estimated) of an underlying linear model? This is important, because it also relates to Fig 2, where the authors plot r_k (without defining it first), so it is hard to understand what is being done on the first read of the main paper.

In general, the methodological section should be expanded. I do not require full theory details and proofs, and strongly propose to avoid jargon, but a schematic working of MR should be illustrated. This is actually easy, since the implementation of the procedure is simple: you perform k_{\max} linear regressions on the data with different time lags k , and then regress the linear regression coefficients against the m^k model. Authors have the space needed for this in Nature Communications. An accessible explanation plus a schematic would go a long way of clarifying the currently mysterious inner workings of the MR.

2) As mentioned above, please relate to physics literature about “critical slowing down” in dynamical systems, and state in an accessible way that you’ll use the estimate of the timescale of

correlation to infer "m". This explanation is much simpler than the current technical explanation in line 75 and following (which you can also retain if you so wish).

3) Subsampling (line 208): you chose subsets of electrodes (not neurons), although the SI states that activity is defined in terms of spike-sorted neurons. So strictly speaking, what are the units that you "subsample", given that the map between electrodes and neurons is not one-to-one? I am not sure how the subsampling is really done — if you removed an electrode, did you re-do spike sorting?

4) I presume there exist stochastic processes whose autocorrelation function does not decrease monotonically, but, e.g., describes either spontaneously damped oscillations. In this case, you couldn't fit r_k with a power law decay well, yet the "envelope" (damping) could exhibit a long timescale. What happens if you apply your estimation procedure to such cases (presumably this can be realized by higher order AR processes)?

5) Ref 2 mentions retinal data, which was claimed to be close to critical in a very different measure (not related to dynamics), see Proc Nat'l Acad Sci USA 112 (2015): 11508-11513; PLOS Comp Biol 10 (2014): e1003408; PLOS Comput Biol 13 (2017): e1005763. It would be interesting, and biologically relevant, to know what the dynamical estimator by W&P estimates, acknowledging that the "statistical criticality" and "dynamical criticality" are not necessarily one and the same concept. This data is publicly available. I encourage the authors (but don't require them) to take a look and report the results in the publication, given the timeliness of the problem.

6) I would ask the authors to make available their code and the scripts to run it on one of the example datasets. This would strongly increase the impact of the work.

Gasper Tkacik

Reviewer #3 (Remarks to the Author):

Inferring collective states, by Wilting and Priesemann. The authors have been very responsive to the prior round of reviews (in which I participated) and I think there is substantially improved balance between methodological innovation + demonstration with the new insights into disease and neuronal dynamics.

I think the authors misunderstood my own prior concerns about the value of an in silico validation in a simple branching process. As with the authors, I am aware that such models serve essentially as heuristics for more complex processes such as critical dynamics in neural systems. But my point was that the activity of such a model forms a poor ground-truth for validating a method meant to apply to real systems, where there slightly more complex models (e.g. with plasticity) that are known to capture crucial aspects of empirical systems: Validation of the effects of down-sampling in such a model would be a better demonstration of the method because more complex spatial and temporal dependences are present. However, with the additional analyses using the BTW model and edits, the paper has clearly moved on from this issue.

I am thus enthusiastic about the paper and have only minor suggestions on the manuscript:

1. Abstract: I think "fading reverberation" is a very vague/non-descript term for the abstract.
2. P1: The sentence beginning, "In these systems, over-estimating .." is redundant to the following (and better) one. Something about the objective achieved in the paper could then be inserted before the following sentence, "Mathematically, ..."
3. P2: The last word "tremendously" is a bit theatrical.
4. Figure 3d. I do not understand why the error bars are horizontal and not vertical. The x-axis is

not labelled/ticked so I assume the data are simply grouped into monkey/cat/rat and ranked arbitrarily within groups.

5. P4: Again (as per abstract), I do not understand the term "fading reverberation" and not just "damped subcritical" or similar (or better operationalize "fading reverberation" mathematically). Also, to more clearly establish that $m=0.984$ is not critical-like, it would be preferable to plot the CDF's of the amplitude or temporal statistics rather than just an example time series and show their deviation from power law scaling.

6. P4-5, again because criticality and "not criticality" are contested positions in neuroscience, it would be preferable to have some quantitative estimates of the consequences of $m=0.984$: What are the implications regarding the loss of scale-free statistics and power law correlations etc.

Reviewer # 1

Reviewer comment:

This manuscript by Wilting and Preisemann offers a substantially improved estimate of the branching ratio for empirical data. Within the neuroscience community, the topic of criticality is very actively discussed and the last few years have seen improvements in estimating power law distributions, a potential marker for criticality. As the branching ratio is another potential marker (if $m = 1$, the system is critical), this manuscript is of similar importance. However, this manuscript offers more because it shows clearly for the first time the status of the data: these networks are actually not critical. The branching ratio, m , is slightly less than one, indicating that the collective dynamics will not show divergences as expected in a truly critical system. Thus, this manuscript is not just a new method, but a new result. I am not a specialist in epidemiology, but I could imagine that this work would offer improvements in that area as well. The authors have written a very accessible introduction that clearly communicates the general nature of this problem. I found it easy to read, especially since the main results are reported directly and the more detailed math was placed in the supplementary information. The supplementary information will allow specialists to follow the steps in the derivation and to check the other technical points. Overall, this is very important work and I see no reason why it should not have been published already. I read over the previous reviews and the authors' responses to them. Given that it has already been through one round of improvements, I could not find too many suggestions for it. However, I do have some points for the authors to consider:

1. Not critical and not asynchronous: The lines 204-207 seem to state that the activity is neither asynchronous intermittent (AI, $m = 0$) or critical ($m = 1$), as if in vivo activity is equally distant from both states. But a person reading this might think that it is closer to critical than to AI. Numerically this is clearly the case as $(0.98 - 0)2$ is larger than $(0.98 - 1)2$. It is fine to claim that it is neither, but it just seems like this deserves a bit more explanation as the numerical differences are not the same. While this is brought up again in lines 222-227, the differences in collective dynamics are not explained much, other than a qualitative figure (3e). For example, you could talk about optimality in some sense (e.g., information transmission) and measure how much that differs from a system with $m = 0.98$ from a system where $m = 0$ and a critical system where $m = 1$. By this type of measure, in vivo activity seems much closer to critical than AI, I would think. I realize that this could be a fuzzy, nonquantitative area, but effort to quantify some of the typical features advertised in critical systems could help here.

We thank the reviewer for pointing that the distinction between AI, reverberating, and critical systems deserves more elaboration. We now include an additional panel (Fig. 3d) to show exemplarily that avalanche size distributions in cat visual cortex differ from power law scaling, which is a typical approach to identifying criticality in neural data. We now write (ll. 257ff):

"Furthermore, the distributions of avalanche sizes clearly differ from the power-law scaling expected for critical systems, but are well captured by a matched, reverberating model (Fig. 3d)."

Indeed, the implications of our findings regarding AI and critical neuronal dynamics deserve a thorough analysis, which is now possible with our new estimator. However, in this paper the full discussion would distract from the more general problem of subsampling and our novel solution. We therefore prepared a second manuscript dedicated to a neuroscience readership, which addresses our neuroscience results in depth.

Reviewer comment:

2. Spontaneous background activity: I realize that you want to keep the analysis tractable to see how far you can go, and only later add other parameters (as you explain in lines 261-264). Yet a major part of the critical/not critical debate concerns background spontaneous activity. Strictly speaking, a system can only be self-organized critical (SOC) if there is separation of timescales – the relaxation time of the system is much less than the interval between external driving events. This is unrealistic in the brain, though, as there is spontaneous (undriven) activity in neurons. The addition of spontaneous background activity can cause typical measures of criticality like the susceptibility to no longer diverge. This can make the network look slightly subcritical if viewed from the traditional branching model perspective. A different perspective would be that the network is quasi-critical and for each level of background activity there is a different optimum point. This set of points leads to a Widom line. Such a conception is different from stating that the network is slightly sub-critical. The Widom line view has a set of testable predictions that differ from those of the subcritical view. I realize that it is not the purpose of your paper to discuss this. My suggestion is that framing things merely in terms of critical/subcritical may obscure another important dimension to this research area.

We thank the reviewer for raising this perspective on driven network activity. In fact, we developed the novel estimator – besides its robustness to subsampling – precisely to incorporate ongoing activation from external input or

spontaneous activation (in contrast to avalanche size analysis, which is well-defined only for systems with separation of timescales). We now discuss that the observed difference from $m = 1$ might in principle result from finite system size in combination with input, which bounds the maximum attainable m to quasi-critical dynamics on a Widom line. We now write (ll. 272ff):

”Still, additional factors like input or refractory periods may limit the maximum attainable m to quasi-critical dynamics on a Widom line [55], which could in principle conform with our results.”

Reviewer comment:

Minor: You mention the time bins that you used, and they seem reasonably motivated (the time it takes for activity to propagate from one neuron to another). Have you checked the results for robustness with respect to some variation in the bin widths? This could be relevant, as sub-sampling neurons that are further apart should call for longer time bins, as propagation of activity takes longer for longer distances.

We had indeed checked for the recordings that m scales as theoretically predicted $m(\text{bs} = k\Delta t) = m(\text{bs} = \Delta t)^k$ for multiples of the used bin size. Thus indeed m depends on the assumption of the propagation time, but not on the sampling density. In contrast, the autocorrelation times are independent of the bin size. We now point out this information in the description of binning in the supplementary material (ll. 795f):

”Note that m scales with the bin size (bs) as $m(\text{bs} = k\Delta t) = m(\text{bs} = \Delta t)^k$, while the corresponding autocorrelation times are invariant under bin size changes.”

Reviewer comment:

Very minor: In the caption to figure 1, the term MR is used before it is explained as multistep regression three sentences later.

Corrected as suggested.

Reviewer comment:

Line 552: data set -> data set

Corrected as suggested.

Reviewer comment:

In supplementary section 5, you introduce the letter H to describe a set of hypotheses. This was somewhat confusing initially as you had previously used H for another purpose (line 423).

We thank the reviewer for pointing out this double-usage. We have changed the definition of branching processes in the supplementary such that the laws \mathcal{Y} and \mathcal{H} now use scripted font and are clearly distinct from the hypotheses (ll. 478ff).

Reviewer # 2

Reviewer comment:

The central idea of W&P is to determine a crucial parameter “ m ” of an autoregressive process in an unbiased fashion even when the full activity of the system is spatially subsampled; this parameter determines if a system is strongly sub-critical, “reverberating” or near critical, or really critical. The authors show that estimating “ m ” by means of linear regression between two consecutive samples of temporal activity (as is claimed to be traditional) of the system is strongly biased, if the system is undersampled spatially. They propose a MR estimator that, instead of obtaining “ m ” from the amplitude in the system’s response, looks at the decay timescale of activity. They show analytically for AR processes that a MR estimator based on the timescale is unbiased, and validate this with numerical simulations both for more complicated synthetic models as well as on real data.

I have read (and, I believe, understood) the paper. I have also read the referee reports, as I was asked to do, from when the previous manuscript was submitted to Nature. I strongly recommend this work be published in Nature Communications. Specifically, I disagree with the provided report of referee 2, who is critical about the “biological results and interpretation”.

To support my argument that this work deserves publication due to its relevance to many fields, not limited to biology, it is important to make a few general statements first:

*) Neither Nature, or Nature Communications, are not exclusively biological journals — or so we are led to understand by their claims to interdisciplinarity — and therefore not every result needs to be judged just in light of “biological relevance”, which Ref 2 seems to think is the ultimate publication measure. Having said that, I claim that even when the work of W&P *is* judged for biological relevance, it deserves publication (either in Nature or Nature Comms). Both disease spread and neural network activity have been studied in detail in highly sophisticated as well as simple models, and it is still debated what the data is telling us about the qualitative regime of operation of such models. For instance, in neural networks, the dispute is still ongoing about “critical” vs “asynchronous” regimes of activity. W&P give us a quantitative tool not only to decide those two (somewhat artificially imposed) categories, but discuss the whole quantitative range in between them, including the range of m close to but not equal to 1 (reverberating activity). Being a physicist, a real, biological system tuned to precisely the value of 1.0000 makes no sense for me (personally); but whether it is close to 1 or close to 0 makes sense, and W&P provide a simple, tractable tool to address this question. This is clearly biologically relevant, given the amount of published papers (also in high profile journals) hypothesizing one regime vs the other, while their statistics may fall short of correctly and convincingly identifying one or the other regime.

*) The authors do not claim that all the complex processes that motivate their question are true instances of simple autoregressive processes / branching processes, as some previous comments of the referees seem to imply. Reality is more complicated than that, but (i) to define a “gold-standard” synthetic benchmark against which an estimator can be judged and (ii) by acknowledging that simple models often *are* used (by other researchers) to describe complex processes, sometimes due to experimental / data limitations, and so it is relevant how the parameters of those simpler models are estimated from data; (iii) to provide analytical insights and even define precisely the quantities of interest (e.g., “ m ”); it is perfectly valid that W&P treat the simple and perhaps biologically not-fully-realistic cases, such as PAR models. The paper should therefore not be judged as making claims about what mathematical model is appropriate or not for the diverse set of applications (which is anyway a research topic in its own right) but rather as discussing a statistical estimation problem in a given class of models that, if not done correctly, can lead to biologically qualitatively wrong results. It is correct to ask of authors for numerical evidence that the estimator performs well also out of the PAR-class that motivates it, and I believe they have done a good job at that.

*) One should also separate the claims to novelty of long “memory” in close-to-critical systems, and using these specific claims to derive an unbiased estimator (i.e., a statistical procedure to be applied to finite data of a real system). The fact that timescales can diverge close to criticality in many systems have been well known, and even more so than in neuroscience (that Ref 2 alludes to), this has been studied “to death” in physics, in a phenomenon known as “critical slowing down”. The authors currently phrase the paper in a statistical context (where critical slowing down might have been underappreciated?), and I would ask them, given the broad audience, to properly cite the physics literature where slowing down has been studied experimentally, theoretically, and realized to be a crucial technical obstacle when sampling (e.g., from MCMC). While the authors should acknowledge all this previous work, this doesn’t detract from their contribution, i.e., to devise a new (MR) estimator that uses the timescale of slowing down to infer “ m ”, closeness to criticality, in a way that is independent of the degree of subsampling.

**) Disclaimer: the only statement of the authors that I cannot evaluate is whether the use of autocorrelation time to infer “ m ” is really novel in the statistics literature; while I don’t know of such previous work doesn’t mean it hasn’t been done, given that my background is in physics.*

Having made these remarks and while being strongly supportive of this paper, I think an interdisciplinary theoretical contribution can make the bigger impact the more clearly it is explained and the bigger audience can unambiguously understand it. To this end I ask the authors to:

1) Clarify the actual functioning of the estimator and carefully define the quantities they discuss in the main text. Specifically, the authors are unclear around lines 51/52, and also around lines 75 etc. I only understood the MR estimator after reading the SI. Specifically, what is r_1 , what is \hat{m}_C ? Also lines after 75: what is meant by “all regressions r_k ”? Is “regression” here a procedure (linear, nonlinear or whichever), or a particular estimator for the slope (i.e., not the method, but a parameter to be estimated) of an underlying linear model? This is important, because it also relates to Fig 2, where the authors plot r_k (without defining it first), so it is hard to understand what is being done on the first read of the main paper.

In general, the methodological section should be expanded. I do not require full theory details and proofs, and strongly propose to avoid jargon, but a schematic working of MR should be illustrated. This is actually easy, since the implementation of the procedure is simple: you perform k_{max} linear regressions on the data with different time lags k , and then regress the linear regression coefficients against the m^k model. Authors have the space needed for this in Nature Communications. An accessible explanation plus a schematic would go a long way of clarifying the currently mysterious inner workings of the MR.

We agree that the functioning of the estimator deserves better elaboration. We revised the introduction of the conventional estimator as follows (ll. 49ff):

”A conventional estimator \hat{m}_C of m uses linear regression of activity at time t and $t + 1$, because the slope of linear regression directly returns m owing to the autoregressive representation in Eq. (1).”

We then define our inclusion of subsampling (ll. 54ff):

*”To derive the bias quantitatively, we model subsampling in a generic manner in our stochastic framework: We assume only that the subsampled activity a_t is a random variable that *in expectation* it is proportional to A_t , $\langle a_t | A_t \rangle = \alpha A_t + \beta$ with two constants α and β (Supp. 3).”*

Finally, we are more detailed with terminology in the definition of MR estimation. We now write (ll. 75ff):

*”Instead of directly using the biased regression of activity at time t and $t + 1$, we perform multiple linear regressions of activity between times t and $t + k$ with different time lags $k = 1, \dots, k_{max}$. These return a collection of linear regression slopes r_k (note that r_1 is simply the conventional estimator \hat{m}_C). Under full sampling, one expects an exponential relation $r_k = m^k$ (Theorem S2). Under subsampling, however, we showed that all regressions slopes r_k between a_t and a_{t+k} are biased *by the same factor* $b = \alpha^2 \text{Var}[A_t] / \text{Var}[a_t]$ (Theorem S5).*

[...]

However, because b is constant, one does not need to know b to estimate \hat{m} from regressing the collection of slopes r_k against the exponential model $b m^k$ according to Eq. (2).”

The following paragraph then directly treats the relation the autocorrelation time (see comment 2).

Reviewer comment:

2) As mentioned above, please relate to physics literature about “critical slowing down” in dynamical systems, and state in an accessible way that you’ll use the estimate of the timescale of correlation to infer “ m ”. This explanation is much simpler than the current technical explanation in line 75 and following (which you can also retain if you so wish).

We now state that we use an estimate of the autocorrelation time more clearly and indicate that close to $m = 1$ our estimator therefore makes use of critical slowing down. The whole paragraph now reads (ll. 93ff):

*”In fact, MR estimation is equivalent to estimating the autocorrelation time of subcritical PARs, where autocorrelation and regression r_k are equal: We showed that subsampling decreases the autocorrelation *strength* r_k , but the autocorrelation *time* τ is preserved. This is because the system itself evolves independently of the sampling process. While subsampling biases each regression r_k by decreasing the mutual dependence between subsequent observations (a_t, a_{t+k}) , the temporal decay in $r_k \sim m^k = e^{-k \Delta t / \tau}$ remains unaffected, allowing for a consistent estimate of m even when sampling only a single unit (Fig. 1d). Particularly*

close to $m = 1$ the autocorrelation time $\tau = -\Delta t / \log m$ diverges, which is known as critical slowing down. Because of this divergence, MR estimation can resolve the distance to criticality in this regime with high precision.”

Reviewer comment:

3) *Subsampling (line 208): you chose subsets of electrodes (not neurons), although the SI states that activity is defined in terms of spike-sorted neurons. So strictly speaking, what are the units that you “subsample”, given that the map between electrodes and neurons is not one-to-one? I am not sure how the subsampling is really done – if you removed an electrode, did you re-do spike sorting?*

We thank the reviewer for pointing this out. In fact, we chose subsets from the 50 single units in the recording. We now corrected this in the manuscript and write (ll. 237):

”We chose random subsets of n neurons from the total of 50 recorded sorted single units.”

and consistently write ”neurons” throughout the rest of the paragraph.

Reviewer comment:

4) *I presume there exist stochastic processes whose autocorrelation function does not decrease monotonically, but, e.g., describes either spontaneously damped oscillations. In this case, you couldn’t fit r_k with a power law decay well, yet the “envelope” (damping) could exhibit a long timescale. What happens if you apply your estimation procedure to such cases (presumably this can be realized by higher order AR processes)?*

There are two possible scenarios: (i) The autocorrelation function is of the form $\cos(\omega t) \cdot \exp(-t/\tau)$. Then, MR estimation will return a very small τ , which decays close to zero up to the first root of the cosine. In this case, the included test will indicate that MR estimation is invalid. If wanted, possible extensions of MR estimation could of course pre-process the r_k to obtain the envelope of the modulation. (ii) There is a residual oscillation on top of the exponential decay, $r_k \sim \exp(-t/\tau) + \cos(\omega t) \exp(-t/\tau_{\cos})$. In this case, MR estimation returns an exponential which approximately fits the ”roots” of the cosine and retrieves the timescale τ .

From analyzing other data sets, we found that if oscillations are present they are typically described by the second model and τ can be well estimated.

Reviewer comment:

5) *Ref 2 mentions retinal data, which was claimed to be close to critical in a very different measure (not related to dynamics), see Proc Nat’l Acad Sci USA 112 (2015): 11508-11513; PLOS Comp Biol 10 (2014): e1003408; PLOS Comput Biol 13 (2017): e1005763. It would be interesting, and biologically relevant, to know what the dynamical estimator by W&P estimates, acknowledging that the “statistical criticality” and “dynamical criticality” are not necessarily one and the same concept. This data is publicly available. I encourage the authors (but don’t require them) to take a look and report the results in the publication, given the timeliness of the problem.*

We thank the reviewer for suggesting this analysis. We investigated the proposed data, but found that probably due to the non-stationary input to the retina the results did not pass the employed consistency checks. Therefore, we decided to not include the data in the study.

Reviewer comment:

6) *I would ask the authors to make available their code and the scripts to run it on one of the example datasets. This would strongly increase the impact of the work.*

We are currently developing a toolbox for MR estimation, which is supposed to include convenience functions like FieldTrip support. For the moment, we compiled the core functionality of this manuscript, i.e. branching process simulation and MR estimation, together with an example script. This version will be made publicly available on github until possible acceptance of our manuscript. A corresponding statement has been added to the SI (ll. 803f).

Reviewer # 3

Reviewer comment:

Inferring collective states, by Wilting and Priesemann. The authors have been very responsive to the prior round of reviews (in which I participated) and I think there is substantially improved balance between methodological innovation + demonstration with the new insights into disease and neuronal dynamics.

I think the authors misunderstood my own prior concerns about the value of an in silico validation in a simple branching process. As with the authors, I am aware that such models serve essentially as heuristics for more complex processes such as critical dynamics in neural systems. But my point was that the activity of such a model forms a poor ground-truth for validating a method meant to apply to real systems, where there slightly more complex models (e.g. with plasticity) that are known to capture crucial aspects of empirical systems: Validation of the effects of down-sampling in such a model would be a better demonstration of the method because more complex spatial and temporal dependences are present. However, with the additional analyses using the BTW model and edits, the paper has clearly moved on from this issue.

I am thus enthusiastic about the paper and have only minor suggestions on the manuscript:

1. Abstract: I think “fading reverberation” is a very vague/non-descript term for the abstract.

We clarified the sentence and now write:

”We identify consistently for rat, cat and monkey a state that combines features of both and allows input to reverberate in the network for hundreds of milliseconds.”

Reviewer comment:

2. P1: The sentence beginning, “In these systems, over-estimating ..” is redundant to the following (and better) one. Something about the objective achieved in the paper could then be inserted before the following sentence, “Mathematically, ...”

We thank the reviewer for this elegant suggestion and now write (ll. 20ff):

”However, correct risk prediction is essential to timely initiate counter actions to mitigate the propagation of events. We introduce a novel estimator that allows correct risk assessment even under strong subsampling. Mathematically, [...]”

Reviewer comment:

3. P2: The last word “tremendously” is a bit theatrical.

We agree and now write “severely underestimated” (l. 141f).

Reviewer comment:

4. Figure 3d. I do not understand why the error bars are horizontal and not vertical. The x-axis is not labelled/ticked so I assume the data are simply grouped into monkey/cat/rat and ranked arbitrarily within groups.

The error bars are in fact vertical, but the caps merge for small confidence intervals. We now point out this issue in the figure caption:

”(median $\hat{m} = 0.98$, errorbars: 16% to 84% confidence intervals, note that some confidence intervals are too small to be resolved).”

Reviewer comment:

5. P4: Again (as per abstract), I do not understand the term “fading reverberation” and not just “damped subcritical” or similar (or better operationalize “fading reverberation” mathematically).

We agree that the term fading reverberations was vague. We now simply call the regime ‘reverberating’ and give an explanation for this nomenclature in the manuscript (ll. 223ff):

”This clearly suggests that spiking activity *in vivo* is neither AI-like ($m = 0$), nor consistent with a critical state ($m = 1$), but in a reverberating state that shows autocorrelation times of a few hundred milliseconds. We call the range of the dynamical states found *in vivo* *reverberating*, because input reverberates for a few hundred millisecond in the network, and therefore enables integration of information. Thereby the reverberating state constitutes a specific narrow window between AI state, where perturbations of the firing rate are quenched immediately, and the critical state, in which perturbations can in principle persist infinitely long (for more details, see Wilting & Priesemann [54]).”

Reviewer comment:

Also, to more clearly establish that $m=0.984$ is not critical-like, it would be preferable to plot the CDF's of the amplitude or temporal statistics rather than just an example time series and show their deviation from power law scaling.

6. P4-5, again because criticality and "not criticality" are contested positions in neuroscience, it would be preferable to have some quantitative estimates of the consequences of $m=0.984$: What are the implications regarding the loss of scale-free statistics and power law correlations etc.

We agree that there are better solutions to showing differences from criticality compared to the example time series. Therefore, we included a panel (Fig. 3d) to show exemplarily that avalanche size distributions in cat visual cortex differ from power law scaling, which is a typical approach to identifying criticality in neural data. We now write (ll. 257ff):

"Furthermore, the distributions of avalanche sizes clearly differ from the power-law scaling expected for critical systems, but are well captured by a matched, reverberating model (Fig. 3d)."

We display results for the PDF because CDFs tend to be misleading for power laws with a right cutoff (Yu et al. *PLoS one*, 2014). In addition, we prepared a second manuscript intended for a neuroscience readership, which analyses the implications of our neuroscience results in all detail. We now refer to this second manuscript in the text.

REVIEWERS' COMMENTS:

Reviewer #1 (Remarks to the Author):

I have reviewed this before and I have seen it continually improved. I think the quality is quite high now. The supplementary material is substantial and addresses many of the technical issues that readers may have questions about. I think that the insight that the bias is the same factor over multiple time lags is brilliant and has allowed a breakthrough in estimating the branching ratio. We have already used this approach in analyzing our data, and it has changed the way we think about the issue of criticality. The authors have done a very good job of addressing the reviewers' comments, mine included. I strongly recommend that this be published now.

Reviewer #2 (Remarks to the Author):

I find the manuscript improved in clarity and suitable for publication.

Minor:

- 1) The text starting in line 98 and text starting in line 109 are essentially verbatim copies of each other. I guess the authors wanted to remove one copy?
- 2) Figure 2 caption last sentence: "Both estimatorS return the same..."
- 3) Ref 44 is missing the journal name; ref 54 is not a proper citation (is there an arxiv preprint to be cited)?

Reviewer #3 (Remarks to the Author):

The authors have been responsive to the prior round of reviews and I support the publication of the paper in its current form.

Reviewer # 1

Reviewer comment:

I have reviewed this before and I have seen it continually improved. I think the quality is quite high now. The supplementary material is substantial and addresses many of the technical issues that readers may have questions about. I think that the insight that the bias is the same factor over multiple time lags is brilliant and has allowed a breakthrough in estimating the branching ratio. We have already used this approach in analyzing our data, and it has changed the way we think about the issue of criticality. The authors have done a very good job of addressing the reviewers' comments, mine included. I strongly recommend that this be published now.

We thank the reviewer for this positive assessment.

Reviewer # 2

Reviewer comment:

I find the manuscript improved in clarity and suitable for publication.

We thank the reviewer for this positive assessment.

Reviewer comment:

Minor:

1) The text starting in line 98 and text starting in line 109 are essentially verbatim copies of each other. I guess the authors wanted to remove one copy?

We agree and have merged both paragraphs into one, which now reads

“In fact, MR estimation is equivalent to estimating the autocorrelation time of subcritical PARs, where autocorrelation and regression r_k are equal: We showed that subsampling decreases the autocorrelation *strength* r_k , but the autocorrelation *time* τ is preserved. This is because the system itself evolves independently of the sampling process. While subsampling biases each regression r_k by decreasing the mutual dependence between subsequent observations (a_t, a_{t+k}) , the temporal decay in $r_k \sim m^k = e^{-k\Delta t/\tau}$ remains unaffected, allowing for a consistent estimate of m even when sampling only a single unit (1d). Here, $\tau = -\Delta t / \log m$ refers to the autocorrelation time of stationary (subcritical) processes, where autocorrelation and regression r_k are equal, and Δt is the time scale of the investigated process. Particularly close to $m = 1$ the autocorrelation time $\tau = -\Delta t / \log m$ diverges, which is known as critical slowing down. Because of this divergence, MR estimation can resolve the distance to criticality in this regime with high precision. Making use of this result allows for a consistent estimate of m even when sampling only a single unit (Fig. 1d).”

Reviewer comment:

2) Figure 2 caption last sentence: “Both estimatorS return the same..”

Corrected as suggested.

Reviewer comment:

3) Ref 44 is missing the journal name; ref 54 is not a proper citation (is there an arxiv preprint to be cited)?

Ref [44] was corrected. We now made an arXiv preprint available

Reviewer # 3

Reviewer comment:

The authors have been responsive to the prior round of reviews and I support the publication of the paper in its current form.

We thank the reviewer for this positive assessment.